# Changes in the Level of Premature Mortality in the Polish Population Due to Selected Groups of Cardiovascular Diseases before and during the Pandemic of COVID-19

**DOI:** 10.3390/jcm12082913

**Published:** 2023-04-17

**Authors:** Wacław Moryson, Paweł Kalinowski, Paweł Kotecki, Barbara Stawińska-Witoszyńska

**Affiliations:** 1Department of Epidemiology and Hygiene, Chair of Social Medicine, Poznan University of Medical Sciences, 60-806 Poznan, Poland; 2Department of Hygiene and Epidemiology, Medical University of Lublin, 20-093 Lublin, Poland

**Keywords:** premature mortality, cardiovascular diseases, joinpoint, COVID-19, Poland

## Abstract

Background. In the years 2020 and 2021, the COVID-19 pandemic disrupted Poland’s health care system and caused a high number of excess deaths. After nearly 30 years of continuous dynamic increase in the life expectancy of the Polish population and a decrease in premature mortality that led to a reduction in the health gap between Poland and Western European countries, regrettably, a decline in life expectancy was recorded. For males, the decline amounted to 2.3 years and, for females, to 2.1 years. Aim. The aim of this study was to assess changes in the level of premature mortality due to selected cardiovascular diseases in Poland before and during the COVID-19 pandemic. Method. Time trends of deaths of patients under the age of 65 due to ischemic heart disease, cerebrovascular disease and aortic aneurysm were analyzed by gender and age groups. The joinpoint model was used in determining time trends. Results. Premature mortality due to all of the cardiovascular diseases analyzed had been declining steadily by about 5% per year since 2008. However, at the end of the second decade of the 21st century, a significant change in the dynamics of the trend was observed, particularly with regard to deaths from ischemic heart disease, which since 2018 caused premature mortality increases of 10% per year in the female population. In the male population, an increase of nearly 20% per year has been observed since 2019. The changes also affected premature mortality due to cerebrovascular disease. Conclusions. After nearly three decades of significant decline in premature mortality from cardiovascular diseases in Poland, there was a reversal in the trend, in particular as regards ischemic heart disease. The unfavorable changes intensified in the subsequent two years. The simultaneous increase in the number of cardiovascular incidents ending in death and the decline in access to prompt diagnosis and effective treatment may explain the unfavorable changes in the deaths caused by cardiovascular disease and the increase in premature mortality due to cardiovascular disease.

## 1. Introduction

Until the early 1990s, the level of mortality in the population between the ages of 20 and 64 in Poland had been twice as high as the average for the same age group in the countries of the European Union [1]. Fortunately, the 1990s brought the beginning of a rapid decline in premature mortality. This was a period of dynamic social and political changes in Poland, associated with its political and economic transformation. In the 25 years since the beginning of the Third Republic, Poland became a country in which the values of mortality rates, both from ischemic heart disease and cerebrovascular disease, reached the level of the European average [2]. This positive trend was associated with changes in exposure to risk factors and the introduction of new therapeutic approaches [3]. The years 2008–2017 saw the continuation of a strong downward premature mortality trend in Poland. The level of reduction in premature deaths during this period oscillated around 5% per year, and the total reduction in premature mortality due to cardiovascular diseases amounted to nearly 37%, both among men and women, far exceeding the rate of decline in premature mortality due to cardiovascular diseases for 2013–2025 projected by the World Health Organization (WHO) [4,5]. In 2020, however, the health care system in Poland faced a tremendous challenge due to the COVID-19 pandemic. The first laboratory-confirmed case of COVID-19 in Poland was reported on 4 March 2020 and by 31 December 2021, there were 4,108,216 cases of COVID-19 and 97,054 deaths due to COVID-19, including 22,693 deaths of individuals under the age of 65 [6]. In comparison, the number of premature deaths from ischemic heart disease, i.e., one of the most common causes of mortality in Poland, was 17,424 during this period. Not only did the COVID-19 pandemic in 2020 and 2021 cause a huge number of excess deaths in the country due to SARS-CoV-2 infection, but it also added to the existing health debt that might be attributed to decreased accessibility to hospitals for people with non-communicable diseases. Other factors thereof included excessive occupancy of hospital wards by COVID-19 patients, as well as a limited contact with doctors and delayed diagnosis and treatment of other diseases. A number of studies are now available on changes in premature mortality or life expectancy of a specific population due to COVID-19. The data published by the Central Statistical Office show that between 2020 and 2021, a decrease in life expectancy by 2.3 years for males and 2.1 years for females was observed in Poland [7]. However, there is a lack of studies dedicated to specific non-communicable disease.

This paper presents the results of an epidemiological analysis concerning changes in the level of premature mortality in males and females due to cardiovascular diseases in 2008–2021 in Poland. In the Materials and Methods section the authors describe the methodology used and highlight its advantages and limitations; in the Results section they describe in detail the results of the analysis; in the Discussion section the authors compare their results with the data on the availability of medical services in Poland during the COVID-19 pandemic and the impact of the pandemic on cardiovascular mortality in other countries.

## 2. Materials and Methods

All registered deaths of Polish residents under 65 years of age due to cardiovascular diseases in 2008–2021 were analyzed. The data, regarding the number of deaths and the size of the population in age ranges, come from the Central Statistical Office. The classification of the initial cause of death was made according to the International Statistical Classification, Diseases and Health Problems (ICD-10) [8]. Time trends in mortality were determined for ischemic heart disease (I20–I25), cerebrovascular disease (I60–I69) and aortic aneurysm (I71) (Table 1).

The rows of Table 1 contain diagnosis groups of cardiovascular diseases (ICD-10) included in the analyses and specific diseases within particular groups.

To analyze the phenomenon of mortality in the population of men and women under the age of 65, mortality rates standardized on the basis of the 2013 standard population of Europe were used, as well as crude mortality rates for age groups [9]. The mortality trends were determined for five-year age ranges (20–24, 25–29, 30–34, 35–39,40–44, 45–49, 50–54, 55–59, 60–64 years) with a gender breakdown.

In determining mortality time trends, the joinpoint model, which is a special version of linear regression, was used, representing time trends using a broken curve consisting of segments that connect at points where the change in the time trend is significant. The analysis was carried out with a minimum number of points of change in trend dynamics of zero and a maximum number of one. The significance test of the model containing a point of change in trend dynamics was carried out using the Monte Carlo permutation method. An average annual percentage change in premature mortality due to the analyzed disease groups was calculated using the joinpoint model. It was also determined whether there was a change in the dynamics of the mortality trend over the time period studied. If a year was identified in which a change in the dynamics of the trend was observed, the information on the annual percentage change in mortality before and after that year was provided. Joinpoint Regression statistical software version 4.7.0.0 (US National Cancer Institute) was used to determine the regression model. For both the annual percentage change (APC) and the average annual percentage change in coefficients (AAPC), a confidence interval (CI) of 95% was determined, with *p* < 0.05 used as the level of significance.

Although Joinpoint Regression cannot be used to formulate any conclusions regarding the cause–effect correlation, it was used repeatedly to assess changes in time series data that occur after various interventions or changes in external conditions. To date, this analysis has been used, inter alia, to estimate the changing trends in cancer incidence and mortality [10,11,12], osteoporosis [13], suicide incidence [14] or the changes in traffic-related death rate after the introduction of new traffic regulations [15].

## 3. Results

Between 2008 and 2021, in Poland 441,757 premature deaths in females and 1,064,683 premature deaths in males were recorded. Cardiovascular diseases were responsible for nearly 11% of these deaths in the female population and nearly 15% in the male population, accounting for 47,994 premature deaths in women and 154,273 premature deaths in men.

Ischemic heart disease was responsible for 5.4% of all premature deaths among women and 9.3% among men. Cerebrovascular disease caused 5.2% of all premature deaths in the female population and 4.7% in the male population, respectively, while aortic aneurysm contributed to 0.3% of all premature deaths in women and 0.5% in men (Table 2).

Over the study period. a decrease in premature mortality due to all analyzed groups of cardiovascular diseases was noted. The time trends in most cases followed a similar course. involving a dynamic decline in mortality rates from 2008 to the end of the second decade of the 21st century. Since that time. a change in trend dynamics and the beginning of an increase in the level of premature mortality has been observed (Figure 1).

Premature mortality due to ischemic heart disease decreased in the female population by 4.4% per year between 2008 and 2018. and then began to increase at a rate of 10% per year. Similarly. in the male population. premature mortality declined by 4.6% per year between 2008 and 2019. but then increased by nearly 20% per year. In both the female and male populations. the changes described were statistically relevant (Table 3).

The time trends in premature mortality from cerebrovascular disease followed a similar pattern. Since 2008. a reduction in rates of nearly 6% per year was observed for both sexes. Since 2017. a reduction in the dynamics of the downward trend to 0.5% per year was observed in the female population. On the other hand. in the male population. an increase in premature mortality by as much as 0.4% per year was seen since 2016. The described changes in both genders were also statistically relevant (Table 3).

The trend in premature mortality from aortic aneurysms in the female population was different during the period studied. A constant significant reduction in mortality of 3.7% per year was observed throughout the analyzed period. In the male population. on the other hand. premature mortality decreased significantly by 5.2% per year in 2008–2019 and then began to increase by 6% per year (Table 3).

Premature mortality caused by cardiovascular diseases was decreasing in all studied age groups of both males and females. The trend of mortality in almost all age groups and for all the analyzed diseases. except for aortic aneurysms in females. was divided into a period of rapid decline in mortality rates at the beginning of the analyzed time frame and a period of decreasing dynamics of mortality reduction or even increase in mortality. which fell at the end of the second decade of the 21st century (Table 4).

### 3.1. Ischemic Heart Disease

The largest average annual decline in mortality (AAPC) was observed in the youngest age groups and amounted to 6.5% per year for females and 4.9% for males. respectively. while the lowest decline was observed in the oldest age bracket (60–64) amounting to −0.3% per year for the male and female populations. In almost all age groups. there was a moment of change in the initially strong downward trend in the level of premature mortality in 2018–2019. after which a significant (more than 10% per year) increase in the values of mortality rates was observed in all these age groups (Table 4).

### 3.2. Cerebral Vascular Diseases

The decrease in mortality in both sexes in all age groups was similar to the level for the entire population of people under 65 (5% per year among females and 3.5% among males). In the younger age groups in the female population. no change was observed in the dynamics of mortality reduction. Among women between the ages of 50 and 54. there was a halting and reversal of the downward trend in mortality in 2019. with an increase of 3.6% per year. Among women between 55 and 59. a change in the dynamics of the trend had already been evident in 2015. but it only resulted in a reduction in the dynamics of the declining trend from 6.6% per year to 2.4% per year.

In the male population. a change in the dynamics of the mortality trend was observed in all age groups. In the age brackets between 50 and 64 years of age. it had already occurred in 2016. Among men between 55 and 59 years of age. a decrease in the dynamics of reduction of premature mortality was observed from 6.1% per year to 1.6% per year. In contrast. a slight increase in the values of mortality rates (0.2% per year in the 50–54 age group and 0.7% per year in the 60–64 age group) was observed in 2016 in men aged 50–54 and 60–64 (Table 4).

### 3.3. Aortic Aneurysm

A decrease in the level of premature mortality due to aortic aneurysms was observed in all age groups in both sexes. In the female population. no change in the dynamics of mortality reduction was observed in any age group. Among men. however. in the age groups of 45–49 and 55–59. a halt in the downward trend was observed in 2018 and 2019, respectively. followed by an increase in the value of mortality rates by more than 15% per year (Table 4).

## 4. Discussion

The increase in premature mortality due to cardiovascular disease can be linked to the observed increased risk of cardiovascular incidents during and after COVID-19 infection and. the previously mentioned. inadequate medical care for cardiac patients who did not suffer from COVID-19.

An analysis carried out with UK Biobank data showed an increased risk of cardiovascular incident and cardiovascular death in patients hospitalized by COVID-19. which was highest during the first 30 days after discharge (10-fold higher mortality from ischemic heart disease). but persisted much longer. In contrast. for COVID-19 non-hospitalized patients. only an increased risk of pulmonary embolism was evidenced [13]. Studies conducted on the US population showed an increased risk of incidence and mortality from cardiovascular disease. including both ischemic heart disease and cerebrovascular disease in all individuals who suffered from COVID-19. regardless of whether they were hospitalized for this reason or not [14,15]. It was also found that patients with cardiovascular disease. after contracting COVID-19. are up to 3.9 times more likely to suffer its severe course and have up to 2.7 times higher risk of death than those without cardiovascular diseases [16].

In the extended analysis of premature mortality due to cardiovascular diseases which already included the years of the COVID-19 pandemic in Poland. only the deaths whose initial cause was attributable to these diseases were taken into account. It was shown that a significant turning point in the dynamics of the trend coincided with the onset of the COVID-19 pandemic. after which an increase in the level of premature mortality or a definite reduction in the downward trend was observed. Premature deaths due to aortic aneurysms in the female population were the exception. as the dynamics of the decreasing trend did not change. Currently. there are not many publications devoted to changes in premature mortality due to non-communicable diseases during the COVID-19 pandemic and not as a result of SARS-CoV-2 infection. but they indicate that the phenomenon observed in Poland also occurred in other countries. Thus. it may be concluded that the level of mortality due to other conditions was indirectly affected by this pandemic.

An analysis conducted in the United States compared mortality rates for ischemic heart disease and cerebrovascular disease and showed a steady decline in premature deaths from 2011 to 2019. Nevertheless. the decline was much slower than that seen in Poland. However, compared to 2019, in 2020 the values of the standardized coefficients increased by 4.1% for heart disease and 5.2% for cerebrovascular disease. respectively. The situation of increased mortality caused by these diseases was explained by overcrowding of hospitals with COVID-19 patients and fewer hospitalizations of people with acute cardiovascular incidents. decreased physician accessibility. as well as changes in patient behavior. e.g., non-compliance with medical recommendations [17]. Another study. focused on changes in cardiovascular mortality in different regions of the United States between 2019 and 2020. confirmed the increase in mortality associated with the COVID-19 pandemic. The study additionally highlighted the greater increase in cardiovascular mortality between 2019 and 2020 that occurred in the regions where a higher incidence of COVID-19 was observed [18,19]. According to data provided by the Office for National Statistics (ONS). there was an increase in the number of deaths caused by myocardial infarction (ICD-10 codes I21, I22, I23 were included as the primary cause of death) in England and Wales between 2019 and 2021 from 19,836 to 20,061. The latter figure is subject to change, as it may have been underestimated due to delays in registration [20].

The higher incidence and mortality of cardiovascular disease that further increased in 2020 and 2021 was an indirect effect of the COVID-19 pandemic. which impaired the functioning of the health care system in Poland and changed the health behavior of the Polish people [21,22,23,24,25]. During the pandemic. new impediments to accessing health services became apparent. such as the temporary exclusion of non-COVID patients from health care facilities. A significant percentage of adults in Poland refrained from seeing a doctor not only because of lack of accessibility. but for fear of infection [21]. An analysis on the hospitalization of diabetic patients in Poland showed that during the COVID-19 pandemic there was a marked decrease in hospitalization for diabetes and a higher in-hospital mortality [22]. The authors who studied the quality of care for pediatric patients with type I diabetes in Poland showed that 2020. the first year of the pandemic. brought a 27% decrease in hospitalizations of diabetic patients and as much as a 22% increase in admissions due to ketoacidosis. a life-threatening condition resulting from undiagnosed and untreated or poorly treated diabetes [23]. The number of oncology procedures in the field of urology also declined during this period. The Ministry of Health estimated a decrease in the number of hospitalizations in the field of cardiology in Poland in 2020 by as much as 25%. The number of invasive cardiology procedures performed decreased by approximately 10–20% compared to 2019 [25].

Numerous studies on changes in premature mortality or life expectancy of a specific population due to COVID-19 are currently available. What is lacking. however. are studies dedicated to specific non-communicable diseases [26,27,28]. The analyses concerning chronic non-communicable diseases that used to be key in the hierarchy of morbidity and death. the prevalence of which was adversely affected by the COVID-19 pandemic period. seem particularly valuable.

## 5. Conclusions

The joinpoint method does not make it possible to show a cause-and-effect correlation between increased premature mortality due to cardiovascular diseases and COVID-19 pandemic. However. the above-presented correlations. involving a concurrent increase in the number of cardiovascular incidents ending in death and a decrease in the availability of timely diagnosis and effective treatment. may explain the unfavorable reversal of the trend in premature mortality due to cardiovascular disease and thus a further increase in premature mortality. The increase in the level of premature mortality due to cardiovascular diseases during the COVID-19 pandemic in Poland intensified the already increasing trend. The previous positive trend was reversed and a further slowdown in the dynamics of the downward trend was recorded with respect to cardiovascular diseases.

This unfavorable phenomenon can be linked to the lack of access to timely diagnosis and effective treatment of cardiac patients. as well as to changes in the behavior of Poles due to social distancing and voluntary limitation of contacts with medical facilities for fear of infection.

However. the limitations of this work should be noted. The number of deaths due to cardiovascular diseases during the COVID-19 pandemic was underestimated. The death certificate data for those dying of cardiovascular disease included only those who were not infected with SARS-CoV-2 at the time of death. According to the international rules for registering deaths as provided by the WHO. all deaths involving people with active SARS-CoV-2 infection were classified using the COVID-19 codes. even if it was a cardiovascular disease that contributed to the severity of the SARS-CoV-2 infection. This suggests that had deaths from cardiovascular diseases concurrent with the SARS-CoV-2 infection been included in the register. the increase in premature mortality from cardiac reasons would have been higher. Moreover. it should be emphasized that mortality statistics for cardiovascular diseases in Poland are characterized by their low quality. with almost half of cardiovascular deaths being described using the so-called “garbage codes”.

Not only did the COVID-19 pandemic contribute to numerous excess deaths due to SARS-CoV-2 infection. but it also resulted in inaccessibility to hospitalization for people with non-communicable diseases as well as the excessive occupancy of hospital wards by COVID-19 patients. The pandemic also resulted in limited access to physicians and delayed diagnosis and treatment for patients with other diseases. In view of the above. further analyses examining the impact of SARS-CoV-2 infections and the unfavorable organizational changes related to the COVID-19 pandemic on the incidence and mortality from other diseases seem to be necessary. The proposed research should be also extended to the period beyond the years of the highest number of COVID-19 infections and deaths.

## Figures and Tables

**Figure 1 jcm-12-02913-f001:**
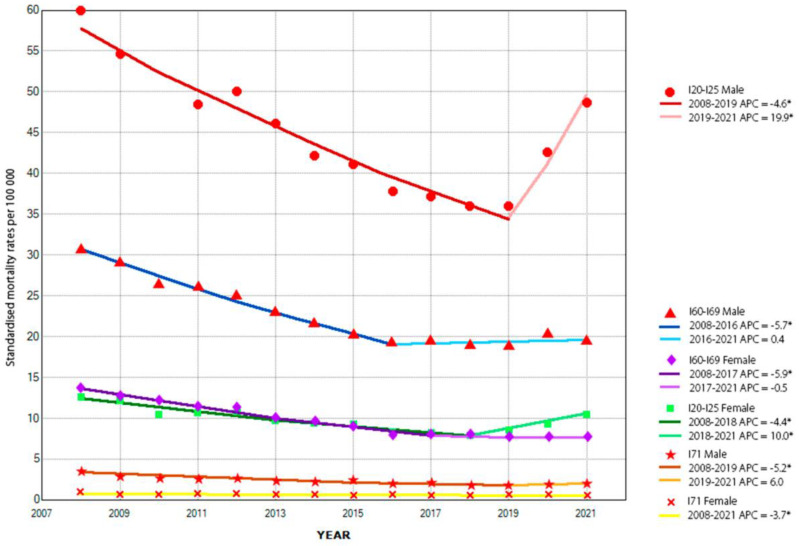
Time trends of standardized mortality rates due to the analyzed groups of cardiovascular diseases. contain information on annual percentage change (APC) of standardized mortality rates due to analyzed groups of cardiovascular diseases. * the significant values were marked at *p* < 0.05.

**Table 1 jcm-12-02913-t001:** The analyzed groups and cardiovascular diseases.

I20–I25	I20	I21	I22	I23	I24	I25				
Ischemic heart disease	Angina pectoris	Acute myocardial infarction	Subsequent myocardial infarction	Complications following acute myocardial infarction	Other acute ischemic heart diseases	Chronic ischemic heart disease				
I60–I69	I60	I61	I62	I63	I64	I65	I66	I67	I68	I69
Cerebrovascular diseases	Subarachnoid haemorrhage	Intracerebral haemorrhage	Other nontraumatic intracranial haemorrhage	Cerebral infarction	Stroke, not specified	Occlusion and stenosis of precerebral arteries, not resulting in cerebral infarction	Occlusion and stenosis of cerebral arteries, not resulting in cerebral infarction	Other cerebrovascular diseases	Cerebrovascular disorders in diseases classified elsewhere	Sequelae of cerebrovascular disease
I71										
Aortic aneurysm and dissection										

**Table 2 jcm-12-02913-t002:** Premature deaths due to groups of cardiovascular diseases in Poland by gender, years 2008–2021.

Year	Females							Males						
	All Premature	I20–I25		I60–I69		I71		All Premature	I20–I25		I60–I69		I71	
		(N)	(%)	(N)	(%)	(N)	(%)		(N)	(%)	(N)	(%)	(N)	(%)
2008	33,409	1977	5.9	2126	6.4	127	0.4	82,794	8671	10.5	4467	5.4	487	0.6
2009	33,968	1981	5.8	2064	6.1	101	0.3	82,203	8114	9.9	4345	5.3	427	0.5
2010	32,782	1799	5.5	2051	6.3	90	0.3	80,761	8228	10.2	4141	5.1	413	0.5
2011	32,897	1851	5.6	1961	6.0	99	0.3	80,232	7715	9.6	4162	5.2	410	0.5
2012	33,118	1897	5.7	1950	5.9	111	0.3	80,076	8023	10.0	3999	5.0	412	0.5
2013	32,553	1727	5.3	1772	5.4	89	0.3	77,361	7412	9.6	3725	4.8	391	0.5
2014	31,124	1672	5.4	1656	5.3	93	0.3	72,877	6784	9.3	3487	4.8	359	0.5
2015	30,991	1634	5.3	1521	4.9	76	0.2	73,777	6630	9.0	3266	4.4	389	0.5
2016	29,910	1468	4.9	1369	4.6	100	0.3	71,855	6051	8.4	3089	4.3	339	0.5
2017	29,470	1453	4.9	1368	4.6	79	0.3	70,797	5941	8.4	3177	4.5	343	0.5
2018	29,696	1389	4.7	1377	4.6	76	0.3	70,750	5710	8.1	3018	4.3	292	0.4
2019	28,604	1469	5.1	1285	4.5	73	0.3	68,683	5701	8.3	2987	4.3	298	0.4
2020	30,206	1589	5.3	1289	4.3	77	0.3	74,297	6622	8.9	3181	4.3	303	0.4
2021	33,029	1771	5.4	1271	3.8	66	0.2	78,220	7442	9.5	3007	3.8	315	0.4
total	441,757	23,677	5.4	23,060	5.2	1257	0.3	1,064,683	99,044	9.3	50,051	4.7	5178	0.5

N—Absolute Numbers. %—Percentage.

**Table 3 jcm-12-02913-t003:** Time trends of standardized mortality rates due to the analyzed groups of cardiovascular diseases.

ICD 10	CARDIOVASCULAR DISEASES	YEARS	Females		YEARS	Males	
			APC (95% CI)	AAPC (95% CI)		APC (95% CI)	AAPC (95% CI)
I20–I25	ISCHAEMIC HEART DISEASE	2008–2021		−1.3 * (−2.5, −0.0)	2008–2021		−1.2 * (−2.4, −0.0)
		2008–2018	−4.4 * (−5.2, −3.6)		2008–2019	−4.6 * (−5.2, −4.0)	
		2018–2021	+10.0 * (+4.0, +16.4)		2019–2021	+19.9 *(+9.9, +30.7)	
I60–I69	CEREBRAL VASCULAR DISEASE	2008–2021		−4.3 * (−5.2, −3.3)	2008–2021		−3.4 * (−4.2, −2.7)
		2008–2017	−5.9 * (−6.7, −5.0)		2008–2016	−5.7 * (−6.6, −4.9)	
		2017–2021	−0.5 (−3.6, +2.6)		2016–2021	+0.4 (−1.5, +2.2)	
I71	AORTIC ANEURYSM	2008–2021	−3.7 * (−5.1, −2.2)	−3.7 * (−5.1, −2.2)	2008–2021		−3.6 * (−6.0, −1.1)
					2008–2019	−5.2 * (−6.3, −4.0)	
					2019–2021	+6.0 (−11.5, +26.9)	

The rows of Table 3 contain information on annual percentage change (APC) and average annual percentage change (AAPC) of standardized mortality rates due to analyzed groups of cardiovascular diseases. * the significant values were marked at *p* < 0.05.

**Table 4 jcm-12-02913-t004:** Mortality trends due to groups of selected cardiovascular diseases in Poland by gender and age.

Age	YEARS	Females		YEARS	Males	
		APC (95% CI)	AAPC (95% CI)		APC (95% CI)	AAPC (95% CI)
ISCHAEMIC HEART DISEASE						
20–24	2008–2021	-	-	2008–2021	-	-
		-	-		-	-
		-	-		-	-
25–29	2008–2021	-	-	2008–2021		−4.9 (−10.3, +0.9)
		-	-	2008–2019		
		-	-	2019–2021		
30–34	2008–2021		−6.5 (−17.0, +5.3)	2008–2021		−1.4 (−5.9, +3.5)
	2008–2012	−35.6 * (−55.7, −6.3)		2008–2014	−15.2 * (−22.3, −7.4)	
	2012–2021	+10.3 (−1.0, +22.9)		2014–2021	+12.3 * (+4.7, +20.3)	
35–39	2008–2021	−1.4 (−6.3, +3.8)	−1.4 (−6.3, +3.8)	2008–2021		−2.3 (−6.0, +1.5)
				2008–2017	−10.8 * (−13.8, −7.6)	
				2017–2021	+19.6 * (+6.0, +35.0)	
40–44	2008–2021		−0.4 (−4.1, +3.5)	2008–2021		−3.5 (−6.8, −0.0)
	2008–2017	−9.1 * (−12.2, −5.9)		2008–2017	−10.9 * (−13.7, −8.0)	
	2017–2021	+22.5 * (+8.6, +38.1)		2017–2021	+15.6 * (+3.4, +29.2)	
45–49	2008–2021	−3.8 * (−6.3, −1.3)	−3.8 * (−6.3, −1.3)	2008–2021		−4.5 * (−6.3, −2.6)
				2008–2019	−7.4 * (−8.3, −6.6)	
				2019–2021	+13.6 (−1.9, +30.3)	
50–54	2008–2021		−2.9 * (−5.7, −0.0)	2008–2021		−2.5 * (−4.3, −0.6)
	2008–2019	−5.8 * (−7.1, −4.5)		2008–2019	−5.7 * (−6.5, −4.8)	
	2019–2021	+14.0 (−6.7, +41.2)		2019–2021	+17.1 * (+2.7, +33.4)	
55–59	2008–2021		−1.2 (−3.4, +1.0)	2008–2021		−1.3 * (−2.5, −0.0)
	2008–2019	−3.5 * (−4.5, −2.5)		2008–2019	−4.3 * (−4.9, −3.7)	
	2019–2021	+12.4 (−3.8, +31.2)		2019–2021	+17.0 * (+6.9, +28.1)	
60–64	2008–2021		−0.3 (−2.1, +1.5)	2008–2021		−0.3 (−1.2, +0.7)
	2008–2018	−3.9 * (−5.1, −2.7)		2008–2018	−3.9 * (−4.6, −3.3)	
	2018–2021	+12.5 * (+3.8, +21.9)		2018–2021	+13.0 * (+8.2, +18.1)	
CEREBRAL VASCULAR DISEASE						
20–24	2008–2021	−1.7 (−6.9, +3.8)	−1.7 (−6.9, +3.8)	2008–2021	−3.9 (−9.1, +1.6)	−3.9 (−9.1, +1.6)
25–29	2008–2021	−0.5 (−3.8, +3.0)	−0.5 (−3.8, +3.0)	2008–2021		+3.5 (−3.3, +10.8)
				2008–2018	−2.7 (−7.2, +2.0)	
				2018–2021	+27.1(−6.1, +72.0)	
30–34	2008–2021	−1.4 (−3.8, +1.0)	−1.4 (−3.8, +1.0)	2008–2021		−1.8 (−5.2, +1.7)
				2008–2011	−14.9 * (−27.2, −0.5)	
				2011–2021	+2.5 * (+0.1, +5.0)	
35–39	2008–2021	−1.9 (−3.8, +0.1)	−1.9 (−3.8, +0.1)	2008–2021		−1.6 (−4.3, +1.2)
				2008–2015	−6.3 * (−10.0, −2.4)	
				2015–2021	+4.3 (−1.0, +9.8)	
40–44	2008–2021	−5.5 * (−7.1, −3.9)	−5.5 * (−7.1, −3.9)	2008–2021		−2.9 * (−4.9, −0.8)
	2008–2017			2008–2017	−6.4 * (−8.2, −4.6)	
	2017–2021			2017–2021	+5.5 (−1.2, +12.7)	
45–49	2008–2021	−5.0 * (−6.8, −3.1)	−5.0 * (−6.8, −3.1)	2008–2021		−3.5 * (−6.5, −0.4)
				2008–2019	−5.6 * (−7.0, −4.1)	
				2019–2021	+8.7 (−13.0, +35.9)	
50–54	2008–2021		−4.9 * (−6.9, −2.8)	2008–2021		−3.5 * (−4.4, −2.7)
	2008–2019	−6.4 * (−7.3, −5.4)		2008–2016	−5.8 * (−6.8, −4.8)	
	2019–2021	+3.6 (−10.7, +20.4)		2016–2021	+0.2 (−1.9, +2.4)	
55–59	2008–2021		−4.7 * (−5.9, −3.4)	2008–2021		−4.4 * (−5.5, −3.3)
	2008–2015	−6.6 * (−8.4, −4.8)		2008–2016	−6.1 * (−7.3, −4.8)	
	2015–2021	−2.4 * (−4.7, 0.0)		2016–2021	−1.6 * (−4.3, +1.1)	
60–64	2008–2021	−4.4 * (−5.4, −3.5)	−4.4 * (−5.4, −3.5)	2008–2021		−2.9 * (−4.1, −1.8)
				2008–2016	−5.1 * (−6.4, −3.9)	
				2016–2021	+0.7 (−2.1, +3.4)	
AORTIC ANEURYSM						
20–24	2008–2021	-	-	2008–2021	-	-
		-	-		-	-
		-	-		-	-
25–29	2008–2021	-	-	2008–2021	−3.9 (−10.3, +2.8)	−3.9 (−10.3, +2.8)
		-	-			
		-	-			
30–34	2008–2021	-	-	2008–2021	−1.8 (−6.5, +3.2)	−1.8 (−6.5, +3.2)
		-	-			
		-	-			
35–39	2008–2021	-	-	2008–2021	−3.4 (−7.7, +1.0)	−3.4 (−7.7, +1.0)
		-	-			
		-	-			
40–44	2008–2021	−1.4 (−10.3, +8.3)	−1.4 (−10.3, +8.3)	2008–2021	−2.0 (−7.1, +3.5)	−2.0 (−7.1, +3.5)
45–49	2008–2021	−3.0 (−9.8, +4.4)	−3.0 (−9.8, +4.4)	2008–2021		−0.6 * (−5.3, −4.3)
				2008–2018	−5.7 * (−8.8, −2.4)	
				2018–2021	+18.3 (−4.7, +46.8)	
50–54	2008–2021	−4.7 * (−9.1, −0.0)	−4.7 * (−9.1, −0.0)	2008–2021	−3.5 * (−5.7, −1.3)	−3.5 * (−5.7, −1.3)
55–59	2008–2021	−6.7 * (−9.4, −3.8)	−6.7 * (−9.4, −3.8)	2008–2021		−3.9 (−7.8, +0.2)
				2008–2019	−7.0 * (−8.8, −5.1)	
				2019–2021	+15.4 (−13.8, +54.4)	
60–64	2008–2021	−2.1 * (−4.2, −0.0)	−2.1 * (−4.2, −0.0)	2008–2021	−4.9 * (−6.2, −3.7)	−4.9 * (−6.2, −3.7)

The relevant rows of Table 4 contain detailed information on annual percentage change (APC) and average annual percentage change (AAPC) in mortality of women and men in the analyzed period by age. * the significant values were marked at *p* < 0.05.

## Data Availability

Data available in a publicly accessible repository that does not issue DOIs Publicly available datasets were analyzed in this study. This data can be found here: http://demografia.stat.gov.pl/bazademografia/Tables.aspx (accessed on 2 December 2022).

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
