# Peer review of "Changes in the Level of Premature Mortality in the Polish Population Due to Selected Groups of Cardiovascular Diseases before and during the Pandemic of COVID-19"

_jcm, 2023, doi:10.3390/jcm12082913_

Round 1

Reviewer 1 Report

In the present study, authors have described the changes in the level of premature mortality in the Polish population due to selected groups of cardiovascular diseases before and during the pandemic of COVID-19. However, authors have analyzed large groups of data to support the conclusion, but significant improvements are needed in order to meet the standard for publication.

1. Manuscript needs extensive English revision. The authors should check the sentence frame throughout the manuscript.

2. Abstract is not clear. Authors should divide the abstract into subheadings.

3. Authors should change the format for Tables 3&4. They are not clear for interpretation. 

4. They have described the enhancement of CVDs mortality during the covid-19 period, but they did not mention how many CVD patients had a covid-19 infection. They should correlate the data between CVD-associated mortality and covid-19 infection.  

5. The discussion is far too long and needs to be focused.

Author Response

Dear Sir/Madam

Thank you very much for your review.

I consider your remarks very valuable. I have analysed them with great diligence and modified the content of the manuscript to implement them. I believe that thanks to the advice provided, the work becomes clearer and more valuable to the readers.

Please find my replies to the passages of the article you have indicated. The changes have also been introduced in the paper.

Manuscript needs extensive English revision. The authors should check the sentence frame throughout the manuscript.

The publication has been linguistically revised.

Abstract is not clear. Authors should divide the abstract into subheadings.

As recommended, the abstract has been divided into paragraphs with appropriate subheadings.  In addition, the content of the abstract has been rewritten to make it more readable.

Authors should change the format for Tables 3&4. They are not clear for interpretation. 

The formatting of Tables 3 and 4 has been changed to make them more legible and readable.

The column containing information on the year in which the change in trend dynamics was observed has been removed from the tables. The column indicating the time interval a certain level of change in premature mortality was observed has been retained. By removing the aforementioned column, the remaining data is presented much more clearly.

Authors have described the enhancement of CVDs mortality during the covid-19 period, but they did not mention how many CVD patients had a covid-19 infection. They should correlate the data between CVD-associated mortality and covid-19 infection.  

In the introduction, we have currently included a comparison of the incidence and death rates and premature deaths from COVID 19 with the number of premature deaths from ischemic heart disease.

The data on deaths from the cardiac diseases analyzed include exclusively those patients who died from ischemic heart disease, stroke or aortic dissection aneurysm and were not currently suffering from COVID 19 infection. In the Discussion section, we cited information that according to the international death coding rules provided by the WHO, deaths from COVID 19 represented the initial cause of death regardless of pre-existing conditions that might have contributed to the severity of COVID-19. In view of the above recommendations, all deaths related to individuals with active SARS COV-2 virus infection were recoded using the COVID 19-appropriate codes. Thus, based on the available data, it is not possible to determine how many of these individuals also suffered from cardiovascular diseases. We also pointed to such information as one of the limitations of our work.

The discussion is far too long and needs to be focused.

The discussion has been rewritten and abbreviated.

Reviewer 2 Report

The introduction presents the topic to the reader and there is some information that provide a background understanding to the reader. However, you need to present what other studies have found on similar topics explored? what does literature indicate, what methods have been employed and what outcomes have been expected? This would allow to show the gap that your work will cover. In addition, the structure of the paper should be at the end of the introduction. Moreover, the material and methods part should include other studies that have employed this technique along with the advantages and disadvantages associated to that.  Overall results are well presented and discussion discusses the important findings based on literature - which would be expected to be explained after the introduction. Conclusions present some of the main highlights however policy implications and recommendations have to be made clear along with the limitations of your work and further research. These points are deemed crucial and have to be developed in your work.

Author Response

Dear Sir/Madam

Thank you very much for your review.

I consider your remarks very valuable. I have analysed them and modified the content of the manuscript to implement them. I believe that thanks to the advice provided, the work becomes clearer and more valuable for the readers.

Please find my replies to the passages of the article you have indicated. The changes were also introduced in the paper.

You need to present what other studies have found on similar topics explored? what does literature indicate, what methods have been employed and what outcomes have been expected?

The introduction has been edited. The information on the current state of knowledge regarding the impact of the COVID 19 pandemic on life expectancy has been added. It has also been pointed out that there currently is a lack of analyses focusing on specific non-communicable diseases.

In addition, the structure of the paper should be at the end of the introduction.

An outline of the publication's structure has been  provided at the end of the introduction.

Moreover, the material and methods part should include other studies that have employed this technique along with the advantages and disadvantages associated to that. 

The Materials and Methods section currently includes the examples of other studies using the Joinpoint method and discusses its advantages and disadvantages.

Overall results are well presented and discussion discusses the important findings based on literature - which would be expected to be explained after the introduction. Conclusions present some of the main highlights however policy implications and recommendations have to be made clear along with the limitations of your work and further research. These points are deemed crucial and have to be developed in your work.

The Conclusions section has been revised and a section on the limitations of the work and areas for further research has been included.

Round 2

Reviewer 2 Report

Thank you very much for considering the comments I have made, I believe the quality of your work has been improved.

Some last points that should be accounted:

- At the end of the introduction you have to present the structure of the whole work - not only the discussion part so the reader will know what are the topics to be explored further.

- At  your conclusion you need to mention the limitations of your work - what did not work well? what has not been considered? This should be before the further research part.

Author Response

Dear Sir/Madam

Thank you very much for your review.

Please find my replies to the passages of the article you have indicated. The changes were also introduced in the paper.

- At the end of the introduction you have to present the structure of the whole work - not only the discussion part so the reader will know what are the topics to be explored further.

As advised, a presentation of the entire structure of the work is included at the end of the introduction.

- At  your conclusion you need to mention the limitations of your work - what did not work well? what has not been considered? This should be before the further research part.

In the Conclusion section we mentioned the limitations of the work. Attention was drawn to the limitation of the statistical method and the limitations resulting from the data set used - data from death certificates.